# Soil Carbon and Organic Matter Fractions Under Nitrogen Management in a Maize–Soybean–Cover Crop System in the Cerrado

**DOI:** 10.3390/plants15010090

**Published:** 2025-12-27

**Authors:** Douglas Rodrigues de Jesus, Fabiana Piontekowski Ribeiro, Raíssa de Araujo Dantas, Maria Lucrécia Gerosa Ramos, Thais Rodrigues de Sousa, Ana Caroline Pereira da Fonseca, Heloisa Carvalho Ribeiro, Rayane Silvino Maciel, Karina Pulrolnik, Robélio Leandro Marchão, Cícero Célio de Figueiredo, Arminda Moreira de Carvalho

**Affiliations:** 1Faculty of Agronomy and Veterinary Medicine, Campus Darcy Ribeiro, University of Brasília-UnB, Brasília 70910-970, DF, Brazil; lucrecia@unb.br (M.L.G.R.); 231105531@aluno.unb.br (T.R.d.S.); 251110323@aluno.unb.br (A.C.P.d.F.); 190088591@aluno.unb.br (H.C.R.); 211052118@aluno.unb.br (R.S.M.); cicerocf@unb.br (C.C.d.F.); 2Embrapa Cerrados, BR 020, km 18, Planaltina 73310-970, DF, Brazil; fbn2.ribeiro@gmail.com (F.P.R.); raissa.dantas@colaborador.embrapa.br (R.d.A.D.); karina.pulrolnik@embrapa.br (K.P.); robelio.marchao@embrapa.br (R.L.M.); 3Department of Forestry Engineering, Campus Darcy Ribeiro, University of Brasília-UnB, Brasília 70910-970, DF, Brazil

**Keywords:** global climate change, no-tillage, low C agriculture

## Abstract

Using cover crops (CCs) following annual crops, together with sustainable nitrogen (N) management, significantly enhances soil carbon (C) storage. However, carbon accumulation in tropical soils is strongly influenced by the respective crop sequences. This study evaluated soil C stocks and fractions in a system incorporating maize–soybean rotation and successive CCs. A randomized block design with split plots was implemented, where main plots consisted of different CCs and the subplots of treatments with and without N fertilization of maize. Chemical fractions of soil organic matter (SOM) were analyzed at depths from 0 to 40 cm, and C stocks down to 100 cm. The SOM fractions responded to N topdressing of maize, varying with soil depth. Soil C stocks during the maize phase were significantly higher than during soybean cultivation (*p* < 0.05), likely reflecting greater residue inputs from species with elevated C:N ratios. Legume crops following maize intensified C accumulation, emphasizing the importance of N inputs for soil C dynamics. Soil C losses were lowest in the treatments with *Raphanus sativus* without and *Crotalaria juncea* with N fertilization. These findings highlight the relevance of combining CCs and N management to optimize C sequestration in tropical agroecosystems.

## 1. Introduction

The agriculture, forestry, and other land use (AFOLU) sector accounts for 74% of all Brazilian greenhouse gas (GHG) emissions [1]. This exemplifies how strongly agricultural land-use systems influence soil carbon stocks in tropical regions. The global SOC pool to 1-m depth is estimated at about 1500 Pg C, of which more than 140 Pg C is found in the top 30 cm of cropland soils [2]. In this context, capture and storage of soil organic carbon (SOC) in croplands came to be recognized as a nature-based solution for carbon dioxide (CO_2_) removal.

As one of the world’s leading agricultural exporters in tropical regions, Brazil depends critically on soybean and maize production [3,4,5]. According to the Brazilian National Supply Company (CONAB), Brazil remains the world’s leading soybean producer. For the 2024/25 growing season, soybean acreage was projected at 47.35 million hectares, with an expected grain output of 171.5 million tons, while maize was forecast to cover 21.86 million hectares and yield 139.7 million tons, consolidating Brazil as the third-largest maize producer globally [6]. Approximately 42% of the country’s agricultural land, particularly for grain production, is located in the Cerrado biome [7]. However, Cerrado soils are highly weathered, nutrient-poor, with low natural fertility. Consequently, crop production in this region depends heavily on external inputs, particularly nitrogen (N) fertilization, which is intensively applied to maximize yields and, consequently, increase crop biomass returns to the soil [8].

One widely adopted strategy to increase soil nitrogen (N) is the use of cover crops (CCs), a practice that has expanded rapidly in Brazilian agriculture, especially across the Cerrado biome. Cover crops can also contribute to building soil organic carbon (C) stocks [9,10]. When combined with N fertilization of the cash crop, CCs deliver multiple benefits: enhanced functional properties of the cover crop itself, higher yields of the commercial crop, and increased C sequestration in the soil. However, the magnitude of these benefits strongly depends on cover-crop species traits, particularly the plant chemical composition and the resulting effects on soil N cycling and availability [11].

Cover crops contribute to soil carbon (C) accumulation through the various functions they perform in the soil. They improve soil health and quality and, consequently, enhance the productivity of subsequent cash crops [12]. In addition, CCs help prevent soil degradation and protect soil organic matter (SOM) [13], stimulate soil biological activity [14,15], and enhance N supply [11,16,17]. Consequently, they increase soil C stocks [18,19,20] through biomass inputs from both shoot and root systems, which often improves even deeper soil layers [15,21].

Cover crops are also used to enhance nutrient availability [22], particularly of N [23,24], for subsequent cash crops. This improvement is a result of biological N fixation (BNF) and the decomposition of crop residues, resulting in the release of essential nutrients [19,25]. All these functions of CCs collectively contribute to SOM accumulation, which occurs in different fractions with distinct functions and residence times in primarily the soil, ranging from days to centuries. Chemical fractionation is a widely utilized technique to assess the impact of cover crops on SOM accumulation. This method analyzes three main fractions: fulvic acids (FAa), humic acids (HAa), and humin (HUM) [22], which are chemically distinguished by their solubility. Humin (HUM) is classified as an insoluble humic substance (HS); humic acids (HAs) are soluble under alkaline conditions; fulvic acids (FAs) are soluble in both acidic and alkaline environments [26]. These specific SOM fractions serve as key, more sensitive indicators of soil quality than total carbon stocks, especially in highly weathered soils such as Oxisols [18].

Humic substances (HSs) are understood as multifunctional natural catalysts in sustainable agriculture. They interact with ions to form complexes of varying stability and structural characteristics, creating new opportunities to improve soil health, crop yield, and environmental resilience [27,28]. Due to these properties, HSs can indicate the adequacy of soil management in agricultural areas [29]. Their complex molecular structure constitutes an abundant C and energy source for beneficial soil microorganisms such as bacteria, fungi, and actinomycetes [30]. In addition, humic fractions enhance particle cohesion, stabilize soil aggregates, influence aggregate-size distribution, and are closely linked to soil C conservation [31,32].

Soil C contents and stocks vary depending on the crop rotation sequence and successive cover crops. In this experiment, soil C stocks increased during the maize phase but declined significantly after the transition to soybean in 2021, except in the sorghum (*Sorghum bicolor*) and wheat (*Triticum aestivum*) treatments [33]. We hypothesized that maize–soybean rotations combined with CCs improve N availability and biomass production, thereby increasing soil C stocks. To this end, this study analyzed soil C contents and stocks in 2018 and 2024, assessed changes in soil C delta (ΔC), and determined maize and soybean yields alongside CC biomass/dry matter production in the same years. Additionally, humic fractions of SOM in a long-term experiment, under maize treated with and without N topdressing, were compared in 2024.

## 2. Results

### 2.1. C and N Input from Cover Crop Biomass

No significant differences were observed among treatments with regard to C and N inputs from cover crop biomass in 2017. In the comparison between N management practices (WN and NN), C and N inputs were highest in the treatment with *Mucuna aterrima* when N maize topdressing had been applied (Table 1).

In 2023, C inputs were higher in the *Raphanus sativus* and *Cajanus cajan* treatments compared to *Crotalaria juncea* under WN management (residual effect of maize cultivation). Under NN management, C input was higher in the *Cajanus cajan* treatment than under *Raphanus sativus* and *Crotalaria juncea*. No significant differences in C input were observed between N management practices. Nitrogen inputs were higher from *Mucuna aterrima* and *Raphanus sativus* than from *Crotalaria juncea* in subplots with maize N topdressing. In the NN subplots, *Cajanus cajan* provided higher N input than *Crotalaria juncea* (*p* < 0.05). Again, no differences in N input were detected between N management practices (Table 2).

### 2.2. Total C Content, C Stocks, and Soil Organic Matter Fractions

Soil C contents measured in different treatments, years, and depths in maize–soybean rotation followed by CCs are presented in Figure 1. In 2018 (maize phase), total soil C contents were not affected by N management at any depth (*p* > 0.05). Across treatments and N management practices, the C contents in *Cajanus cajan* and *Crotalaria juncea* systems were highest in plots without N application in the 60–80 cm layer (*p* < 0.05) (Figure 1a,b). In 2024 (end of soybean phase), C contents were higher in the *Mucuna aterrima* than *Cajanus cajan* system in the 20–40 cm layer of plots with N fertilization (*p* < 0.05) (Figure 1c,d). At this stage, no significant differences were detected between N management practices at any evaluated depth (*p* > 0.05).

Total soil C stocks (Mg ha^−1^) to a depth of 100 cm in the different years and treatments are listed in Table 3. In 2018 and 2024, no significant differences were observed between treatments or N managements. However, across years, total C stocks were higher in 2018 in all treatments, regardless of N application (*p* < 0.05).

The contents of humic fractions (FA, HA, and HUM), analyzed in soybean rotation after CCs in 2024, are presented in Table 4. In plots without N application, humic fractions did not differ significantly among treatments at any depth. In plots with N fertilization, the FA content in the 0–10 cm layer was higher under *Cajanus cajan* than *Mucuna aterrima* (*p* < 0.05).

Regarding N management, FA contents were higher in plots without N fertilization at 0–10 and 20–40 cm, except for soybean rotation after *Cajanus cajan*. For HA, contents were highest in N-fertilized plots at 10–20 and 20–40 cm, except for soybean rotation after *Cajanus cajan* (*p* < 0.05). The HUM fraction was highest in the soybean–*Cajanus cajan* plot without N application at 20–40 cm (*p* < 0.05).

### 2.3. Variations in C Stocks (ΔC)

Variations in soil C stocks down to 100 cm in response to different treatments and N managements between 2018 and 2024 are presented in Table 5. In the treatment comparison, without N fertilization, C was least reduced under *Raphanus sativus*, which differed statistically from the other species (*p* < 0.05). In N-fertilized subplots, C stocks decreased more under *Crotalaria juncea* than *Mucuna aterrima* and *Cajanus cajan*. Across N managements, ΔC values for *Mucuna aterrima* and *Raphanus sativus* were higher in the subplots without N (*p* < 0.05).

### 2.4. Crop Yield (Maize and Soybean) and Cover Crop Dry Matter Production

Maize yield (2018/2019) and soybean yield (2023/2024) are shown in Table 4. In 2018/2019, maize yield was higher under *Crotalaria juncea* and *Mucuna aterrima* than under *Raphanus sativus* in the absence of N fertilization (*p* < 0.05). With N application, maize yield did not differ among treatments. Across all treatments, maize yield was higher with N fertilization than without it (*p* < 0.05). In 2023/2024, no significant differences in soybean yield were observed among treatments or between N management practices (Table 6).

Dry matter production of cover crops in 2017 did not differ between plots with and without N fertilization (*p* > 0.05). When comparing N managements, only *Cajanus cajan* produced more biomass with N fertilization (*p* < 0.05) (Table 7). In 2023, *Cajanus cajan* produced more biomass than *Crotalaria juncea* in plots without N application (*p* < 0.05). Regarding the residual effect of N fertilization, *Raphanus sativus* and *Cajanus cajan* resulted in higher dry matter production than *Crotalaria juncea* (*p* < 0.05), and no significant differences were observed between N management practices.

### 2.5. Relationship Between Carbon and Plant Variables

Principal component analysis (PCA) was used to explore relationships among soil C stocks, SOM fractions (FA, HA, and HUM), and plant variables (soybean yield and cover crop dry matter). PC1 and PC2 explained 42.3 and 27.6% of the total variance, respectively, accounting for 69.9% of the overall variability (Figure 2). According to variable contributions, three main groups were identified: (1) HA; (2) C stocks, biomass, and yield; and (3) FA and HUM (Figure 2a). The distribution of treatments with (WN) and without (NN) N fertilization in relation to the evaluated variables is shown in Figure 2b. Clustering indicated that N fertilization of maize (residual effect on soybean) may influence yield and other studied variables. The overlapping ellipses indicated similar behavior among *Crotalaria juncea*, *Cajanus cajan*, *Mucuna aterrima*, and *Raphanus sativus* (Figure 2c).

## 3. Discussion

### 3.1. Soil Carbon

Across the studied systems, soil C concentration was highest in the surface layer (0–5 cm), with decreases ranging from 17 to 37% down to 20 cm and from 54 to 64% down to 100 cm. Regardless of management practices, cropping systems, or region, this vertical gradient persisted [34]. This pattern reflects the vertical heterogeneity of soils, strongly influenced by microbial activity, as both the composition and functions of microbial communities vary along the soil profile, with concentrated microbial activity in the topsoil layers (0–20 cm) of agricultural soils [35,36].

Cover crops such as *Crotalaria juncea* and *Cajanus cajan*, cultivated during the maize phase, have deep taproots that may contribute to greater organic matter accumulation throughout the soil profile [37]. The root exudates of these legume species, primarily composed of organic and amino acids, enhance soil organic C [38]. During the soybean phase (2021–2024), *Mucuna aterrima* was notable for its higher N content in tissues and higher lignin:N ratio [11], which influences residue decomposition dynamics and promotes increased soil C storage.

Residue inputs during the 2018 maize phase, characterized by a high C:N ratio, enhanced soil C stocks, compared to those recorded in 2024. In soybean–CC systems, introducing a CC with a high C:N ratio is essential to prevent C stock depletion [25]. Conversely, CCs with low biomass yield and low C:N ratio accelerate residue decomposition, resulting in lower soil C levels [25]. Following maize, legume cover crops are more effective in increasing soil carbon stocks, highlighting the role of N inputs in balancing the C:N ratio and fostering microbial mineralization [39,40].

The variations in soil C stocks observed in this study revealed distinct accumulation and loss patterns across crops and systems, reflecting the temporal dynamics of soil carbon [41]. These variations promote N mineralization in the soil and the release of carbon (C) in the form of CO_2_ into the atmosphere [42]. However, N availability is essential to sustain high biomass production [43], enhancing photosynthesis and thereby increasing C inputs to offset losses and favor soil C accumulation along the profile [44].

In the system without N fertilization, the minor C reduction under *Raphanus sativus* was likely due to the rapid early growth and high leaf area index, which improves light interception and photosynthetic activity [45], thereby reducing C depletion. In contrast, the system with *Crotalaria juncea*, characterized by a longer growth cycle and greater initial nutrient demand, resulted in higher nutrient export and immobilization in plant tissues [11,46]. Enhanced SOM mineralization, stimulated by N availability through topdressing, intensified microbial decomposition. Consequently, the net C incorporation efficiency of this species was greater, since N application promoted decomposition, mineralization, biomass production, and soil C accumulation [25].

### 3.2. Humic Substances

Nitrogen topdressing applied during the maize phase until 2020, with residual effects on soybean at the end of the experiment, induced reduced fulvic acid (FA) levels in CC treatments (*Mucuna aterrima*, *Raphanus sativus*, and *Crotalaria juncea*), indicating reduced residue cycling in response to N fertilization [18]. This pattern was evident both in the surface layer and down to 40 cm. The FA fraction with low molecular weight and high oxygenation is easily mineralizable, which favors microbial activity in comparison with HA, with a heavier molecular weight [47].

Rapid SOM mineralization, enhanced by N availability in the system, intensifies microbial decomposer activity [41]. Additionally, high levels of labile components, e.g., with high hemicellulose and low lignin:N ratios, facilitate rapid microbial conversion into soil products [25,48].

Nitrogen topdressing during the maize phase increased HA levels, except in soybean–*Cajanus cajan* plots. This increase is an indicator of soil quality, and may indicate improvements in soil profile quality, particularly in deeper layers [49]. Depending on the cover crop species, accelerated mineralization of plant residues in the presence of nitrogen favored straw degradation [18]. Talbot and Treseder [50] described interactions among lignin, cellulose, and N contents in plant residues; lignin protects cell wall polysaccharides from microbial degradation, while cellulose serves as a substrate for lignin degradation. This may explain why soybean–*Cajanus cajan* plots did not differ in HA levels between the two N management systems, since *Cajanus cajan* has low cellulose and high lignin concentrations [11].

In terms of SOM fractions, the soybean–*Cajanus cajan* system was the most stable, regardless of N management, likely due to the greater accumulation of recalcitrant compounds from residue decomposition [51]. With regard to the HUM fraction, only *Cajanus cajan* was affected by N application. Since HUM is an insoluble fraction, mainly composed of aliphatic C, long-chain fatty acids, and esters [18,52], this legume accounted for the highest contents.

### 3.3. C and N Inputs from Cover Crops, Maize–Soybean Yield, and Cover Crop Biomass

For *Mucuna aterrima* in 2017, the N biomass content allowed pre-decomposition assessments that identified increases in total N [11]. The structural composition of residues, in particular high lignin contents, can slow down decomposition and mineralization, thereby altering N availability timing and enhancing N retention in soil particulate organic matter [53]. Moreover, inorganic N in the rhizosphere modulates biological N fixation, reducing symbiotic investment and inducing variations in the source of accumulated N [54]. Effects of cover crops on C and N stocks primarily manifest in the long term; short cycles between incorporation and assessment tend to reflect direct effects of maize fertilization, at the expense of the residual benefits of cover crops [55,56].

In 2023, C input from *Raphanus sativus* and *Cajanus cajan* was higher than from *Crotalaria juncea*, due to the C:N ratio and lignin content of the residues; notably, N fertilization did not significantly affect C input [11]. Nitrogen inputs in response to N fertilization were highest under *Mucuna aterrima* and *Raphanus sativus*, while a high symbiotic fixation capacity of *Cajanus cajan* was confirmed by high N inputs even without N fertilization. These results highlight the stability of N cycling by CCs and recommend *Cajanus cajan* as a strategic species for low-fertilizer input systems [57].

Diversified crop management practices with rotation and cover crops can increase N use efficiency, enhancing nutrient uptake and use [58]. Nitrogen fertilization in maize–*Cajanus cajan* treatments significantly increased biomass, underscoring the role of N as a key factor for plant growth and development [43,59].

Plants with high biomass and lignin production, such as *Cajanus cajan* and *Raphanus sativus*, decompose slowly and can be succeeded by soybean, which produces rapidly decomposing residues. This sequence synchronizes nutrient release with crop demand and promotes efficient nutrient cycling [25].

Crop sequences of cover species combined with maize–soybean rotation are known to achieve yield increases of 12 to 30%, depending on regional conditions [60,61]. Nitrogen topdressing of maize has increased yields by 28 to 44% [20]; however, N application does not necessarily induce higher soybean yields and may even lead to yield reductions [62,63]. Cover crops with higher shoot N uptake [11] and faster decomposition rates, e.g., *Crotalaria juncea*, are preferable in maize-associated systems.

### 3.4. Interactions Among the Study Variables

Principal component analysis revealed that soil C stocks correlate directly with biomass production and crop yield, suggesting that agricultural systems and practices promoting biomass accumulation through cover or main crops enhance soil C stocks [20,64]. This relationship of organic matter accumulation and, consequently, enhanced soil C by crop rotation and CC use, is well documented in the literature [18,65,66].

Due to the low C:N ratio of their residues, biological N fixation (BNF) by legumes increases soil N availability [67]. The greater variation in soil C stock observed for *Raphanus sativus*, the only Brassicaceae studied (Figure 2c), may be related to its low lignin concentration and lignin:N ratio (46 g kg^−1^ and 2.45, respectively), which favors rapid residue decomposition [25].

Distinct patterns for soil C fractions (HA, FA, and HUM) were observed (Figure 2). Humic acids diverged from the other groups, which was attributed to their chemical and structural properties, i.e., low mobility [68], high polymerization, and advanced stages of SOM humification [69], indicating greater stability in soils. Unlike fulvic acids, HA is not promptly renewed through residue decomposition. Soil quality and agricultural productivity depend strongly on the availability of humic substances (HSs) such as HUM and HA, which play important roles in the soil biota [18,49].

## 4. Materials and Methods

### 4.1. Experimental Area—Location and Characteristics

The experiment was conducted at Embrapa Cerrados, in Planaltina, Federal District, Brazil (15°35′50.12″ S, 47°42′26.97″ W; altitude 973 m). It was arranged in a randomized complete block design with split plots and three replications, with 12 × 8 m main plots and 6 × 8 m subplots (Figure 3). The main plots represented the CCs *Cajanus cajan* (L.) Millsp. (Fabaceae), *Crotalaria juncea* L. (Fabaceae), *Raphanus sativus* L. (Brassicaceae), and *Mucuna aterrima* L. (Fabaceae), while the subplots corresponded to the application or absence of N topdressing of maize (WN and NN, respectively).

According to the Köppen–Geiger classification, the regional climate is Aw (tropical savannah), characterized by a distinct dry season (winter) and rainy season (summer) [70,71]. During the study period, the average annual rainfall was 1.187 mm and the mean air temperature was 21.73 °C. In 2018, the mean maximum and minimum temperatures were 28.63 and 15.89 °C, respectively, with total annual precipitation of 1349.40 mm. In 2024, the mean maximum temperature reached 29.46 °C and the mean minimum 16.37 °C, while total annual precipitation was 1278.20 mm (Figure 4).

The soil of the experimental site was classified as Oxisol. Soil chemical properties are summarized in Table 8. Soil chemical analyses were performed according to the methods described by Sparks [72].

### 4.2. History of the Study Area, Experimental Design, and Management Practices

Between 1995 and 2005, the area was left fallow. In 2005, an experiment of an annual maize–CC off-season fallow sequence with maize hybrid Pioneer^®^ 30F53VYHR was initiated (Figure 5), which assessed cutting management at flowering and maturity stages [20,25].

In the 2010/2011 growing season, the experiment was subdivided according to nitrogen topdressing of maize (treatments with and without N fertilization, WN and NN, respectively). Hybrid maize 30F53VYHR was sown in November in a no-tillage system (directly on CC residues). Plant density was approximately 65,000 plants ha^−1^, with a row spacing of 0.75 m. At planting, all plots and subplots received 20 kg N ha^−1^, 65.5 kg P ha^−1^, 66.4 kg K ha^−1^, 2 kg Zn ha^−1^ (ZnSO_4_ 7H_2_O), and 10 kg ha^−1^ FTE BR 12 as micronutrient source (containing 3.2% S, 1.8% B, 0.8% Cu, 2.0% Mn, 0.1% Mo, and 9.0% Zn). Nitrogen topdressing of maize of 130 kg N ha^−1^ was applied as urea in two split doses at the phenological stages V4 and V8. Weed control consisted of glyphosate and 2,4D for pre-planting desiccation, atrazine (pre-emergence herbicide), and glyphosate (post-emergence). In the growing seasons 2021/2022 to 2023/2024, soybeans replaced maize as the main crop. Soybean was sown in November in rows spaced 0.5 m apart, at a density of about 220,000 plants ha^−1^. Phosphorus and potassium were applied at 59 and 37.4 kg ha^−1^, respectively, at sowing. To control weeds, glyphosate and glufosinate (for pre-planting desiccation), and glufosinate (for post-emergence desiccation) were applied. The land-use sequence from 1995 to 2024 is illustrated in Figure 5.

After the maize and soybean harvests (March and February, respectively), the CCs were planted in April, without fertilization (under residual effects of commercial crop fertilization) and cut at flowering (between May and August, depending on the plant species). Sowing densities were 40 plants m^−2^ for *Cajanus cajan* and *Crotalaria juncea*; 20 plants m^−2^ for *Mucuna aterrima*; and 80 plants m^−2^ for *Raphanus sativus*. Row spacing for CCs was 0.5 m. Vegetative cycles were as follows: *Cajanus cajan* (70–90 days), *Crotalaria juncea* (90–100 days), *Mucuna aterrima* (140–180 days), and *Raphanus sativus* (45–60 days) [37]. No weed control was applied to the cover crops.

### 4.3. Soil Sampling and Analyses

Soil sampling for carbon stock assessment was carried out in 2018 and 2024, after maize and soybean harvests, respectively, under CCs. Soil was collected from the layers 0–5, 5–10, 10–20, 20–40, 40–60, 60–80, and 80–100 cm. After the soybean harvest in 2024, soil was sampled again to determine SOM fractions (0–10, 10–20, and 20–40 cm deep).

Bulk density was measured in undisturbed samples collected from trenches in native Cerrado at a depth of 120 cm, approximately 50 m away from the experimental site [20]. Duplicate samples per depth were collected from opposite trench walls using stainless-steel rings. Samples were oven-dried at 105 °C and bulk density was calculated based on the ring volume.

Total carbon content was determined in sieved (<2 mm), ground, and re-sieved (<150 µm) soil samples [73,74]. Subsequently, total C was determined using a CHNS elemental analyzer (Macro Vario Cube-Elementar, model 2400 Series II CHNS/O). Carbon stocks were calculated via the equivalent soil mass approach [75], using the following:$$C\;stock=\left(\frac{TC\times BD\times h}{10}\right)$$
where

C stock: in Mg ha^−1^;

TC: total carbon (g kg^−1^);

BD: bulk density (g cm^−3^);

h: soil layer thickness (cm).

The change in soil C stock (ΔC) between 2018 and 2024 was calculated as the difference: 2018–2024.

For SOM chemical fractionation, humic fractions were determined based on the differential solubility method [76].

One gram of soil sample was placed in a Falcon tube with 20 mL of 0.1 mol L^−1^ NaOH. The solution was shaken for 4 h at 80 rpm, left to stand for 12 h, and then centrifuged at 3800 rpm for 30 min. The supernatant was collected, and 20 mL of NaOH was added to the residue, followed by agitation for 2 h 30 min and centrifugation for another 30 min at 3800 rpm. The combined supernatants constituted the alkaline extract containing HA and FA. The remaining insoluble precipitate, identified as humin (HUM), was oven-dried at 50–60 °C.

After centrifugation for 20 min at 3800 rpm, the supernatant (FA) was separated from the precipitate (HA), which was treated with 20 mL of 0.5 mol L^−1^ NaOH, homogenized, and the volume adjusted to 50 mL with distilled water. The C contents of HA and FA fractions were determined following the method of Yeomans & Bremner [77].

Samples were digested at 140 °C with potassium dichromate (0.042 mol L^−1^ for FA and HA; 0.167 mol L^−1^ for HUM) and titrated with ammonium ferrous sulfate (AFS). Humic substances (HSs) were calculated by the following:$$C_{org}=\frac{{(V}_{B}-V_{S})\times N\times 3\times \left(\frac{50}{I}\right)}{m}$$
where

V_B_: AFS volume in mL for blank titration;

V_S_: AFS volume in mL for sample titration;

N: normality of AFS (0.25 N for HUM, 0.033 N for FA and HA);

3: conversion factor of the chemical formula from the dichromate ion (Cr_2_O_7_^2−^) to C;

I: aliquot volume for titration (mL);

m: soil sample weight (g).

The equation expresses oxidizable organic carbon in g kg^−1^ dry soil. The method consists of heating to enhance C recovery.

### 4.4. Cover Crop Biomass, Contribution to C and N Inputs, and Maize–Soybean Yield

Total N in CC shoots was determined by digesting 0.2 g of dried sample at 350 °C for 1 h with 10 mL of HClO_4_:H_2_O_2_ (2:1, *v*/*v*). Digests were diluted (1:6, Milli-Q water) and analyzed by flow injection analysis (FIA, Lachat QuikChem Series 2) using the Berthelot colorimetric method. Carbon input was calculated from CC shoot biomass, based on the IPCC default carbon fraction for herbaceous biomass [78]. Inputs of C and N from cover crop biomass were computed as follows:$$C\;or\;N\;input=\frac{C\;or\;N_{shoot}\times CC\;shoot\;biomass}{1000}$$
where

C or N input: in kg ha^−1^,

C or N in shoots: in g kg^−1^,

CC shoot biomass: in kg ha^−1^.

Maize data correspond to the harvest in March 2018, and soybean yield data to the harvest of February 2024. Two replicates per subplot were sampled, encompassing four 4-m rows for maize and three 4-m rows for soybean. Grain yields were corrected to 13% moisture. Maize–soybean productivity (MSP) was calculated as follows:$$GY=\frac{W}{A}\times 10.000$$
where

GY: soybean/maize grain yield (kg ha^−1^),

W: total weight of harvested soybean/maize grains (kg),

A: subplot area (m^2^).

To monitor the residual effect of CCs, aboveground biomass samples were collected in the previous years (2017 and 2023, respectively), during full CC flowering. A 1 m^2^ steel frame was randomly placed on the soil surface. The plants contained within this 1 m^2^ were cut close to the surface and two samples per subplot were taken and oven-dried at 65 °C to constant weight. Dry matter (DM) was calculated as follows:$$DM=W_{f}-W_{i}$$
where

DM is dry matter (kg ha^−1^),

W_f_: sample weight after drying (kg),

W_i_: sample weight before drying (kg).

### 4.5. Statistical Analyses

For C and N inputs from cover crop biomass, means were compared using Tukey’s test (*p* < 0.05).

Carbon content and stock, organic matter fractions, yield, and biomass data were analyzed using linear mixed models to assess fixed effects of treatments (CCs) and maize N topdressing (with and without N) as subplots. The year was treated as a repeated measure within subplots to account for temporal correlations. In long-term field experiments, observations throughout the same plot are correlated over time; treating year as a repeated measure permits modeling this correlation structure, which improves the estimation of standard errors and the inferential power [79,80]. To formally represent the experimental design, we used the notation b = 1, …, B for blocks, i = 1, …, I for main plot treatments (CC), j = 1, …, J for subplot treatments (N), and t = 1, …, T for years (repeated measures). The experimental unit “subject” for the temporal repetition was considered the subplot (b,i,j), evaluated in different years. The general linear mixed model can be expressed as:$$\gamma_{bijt}=\mu +\propto_{i}+\gamma j+{\left(\alpha y\right)}_{ij}+\tau_{t}+{\left(\alpha \tau \right)}_{it}+{\left(y\tau \right)}_{jt}+{\left(\alpha \gamma \tau \right)}_{ijt}\beta_{b}+\delta_{bi}+S_{bij}+\varepsilon_{bij}$$
where μ is the intercept; α_i_, γj, and (αγ)_ij_, are fixed effects of CCs, N, and their interaction, respectively; τ_t_ is the fixed effect of year; the term ${\left(\alpha y\right)}_{ij}$ corresponds to the interaction between CC and year, capturing how the effect of each cover crop treatment varies over time. Interactions with year were included depending on biological plausibility; β_b_~N(0, σ^2^B) is the random block effect; δ_bi_~N(0, σ^2^W) the whole-plot error (block × CC); S_bij_~N(0, σ^2^S) the random subplot intercept, capturing heterogeneity across experimental units over time; and ε_bijt_ is the within-subject error, assumed to follow a temporal covariance structure.

Model adequacy was checked by graphical inspection of residuals (residuals versus fitted values, Q–Q plots) and formal tests for normality (Shapiro–Wilk), homogeneity of variances (Breusch–Pagan test), linearity, and independence of errors. Estimated marginal means (emmeans) were calculated using the Kenward–Roger method to compute degrees of freedom, enhancing the precision of inferences, especially for small sample sizes [81].

Multiple comparisons among treatments, N rates (with and without topdressing), and years used Tukey’s adjustment [82], controlling the familywise error rate at a significance level of α = 0.05. When necessary, the adjusted means obtained from the mixed model were compared using Sidak’s method, which controls Type I error in multiple comparisons more efficiently than Bonferroni, especially in mixed model-based analyses. When applicable, adjusted means were compared using Sidak’s method [83]. For ΔC, when linear model assumptions were not met, the nonparametric Kruskal–Wallis test (α = 0.05) was applied to compare distributions among treatments. Significant differences were further explored by multiple post hoc comparisons using Dunn’s test or the Kruskal–Wallis multiple comparison procedure (agricolae package).

In addition, a principal component analysis (PCA) examined multivariate patterns and relationships among soil properties (C content and fractions) and crop responses (yield and biomass). The rationale for using PCA was to integrate multiple correlated variables into synthetic axes (principal components), thereby reducing dimensionality and highlighting the main gradients of variation associated with management practices. Prior to the analysis, variables were centered and scaled to unit variance to avoid bias due to differences in measurement units. The components were interpreted based on variable loadings, with PC1 and PC2 representing the main directions of variation. By these multivariate patterns, it was possible to identify which treatments were associated with higher soil C fractions, greater biomass, or improved yield. All analyses were performed in R Core Team, 2025; version 4.3.3 [84], using the following packages: lme4 [85] for fitting mixed linear models; emmeans [86] for adjusted means and multiple comparisons; multcomp [87] for simultaneous tests and groupings, and FactoMineR for PCA.

## 5. Conclusions

In successive maize–CCs systems, nitrogen management of the cash crop influences C and N inputs from cover crop biomass. During the soybean phase, C and N inputs were affected by cover crops and N management. Soil C contents were not significantly affected by maize N fertilization in the maize– or soybean–cover crop rotation. Total soil C stocks were only influenced by the exchange of the main crop maize for soybean, with reductions of 1.4 to 10.4% when not fertilized (NN) and 6.7 to 15.7% under N fertilization (WN) management, when maize was replaced by soybean. *Cajanus cajan* showed the greatest stability in fulvic acid (FA) and humic acid (HA) fractions at all depths in response to N fertilization of the previous maize crop. Under *Mucuna aterrima*, *Raphanus sativus*, and *Crotalaria juncea,* FA fractions without and HA fractions with N fertilization were highest. The lowest soil ΔC loss over five years was observed for *Raphanus sativus* without (−1.82) and *Crotalaria juncea* with N fertilization (−9.69). In addition, N fertilization increased maize yields by 28 to 44%. In the maize phase, in response to N fertilization, biomass increased only in plots with *Cajanus cajan*. In the soybean phase, biomass was influenced only by cover crops; biomass stocks under *Cajanus cajan* were higher than under *Crotalaria juncea*. In conclusion, these results highlight the effectiveness of combining cover crops with nitrogen management to promote low-carbon agriculture in tropical regions.

## Figures and Tables

**Figure 1 plants-15-00090-f001:**
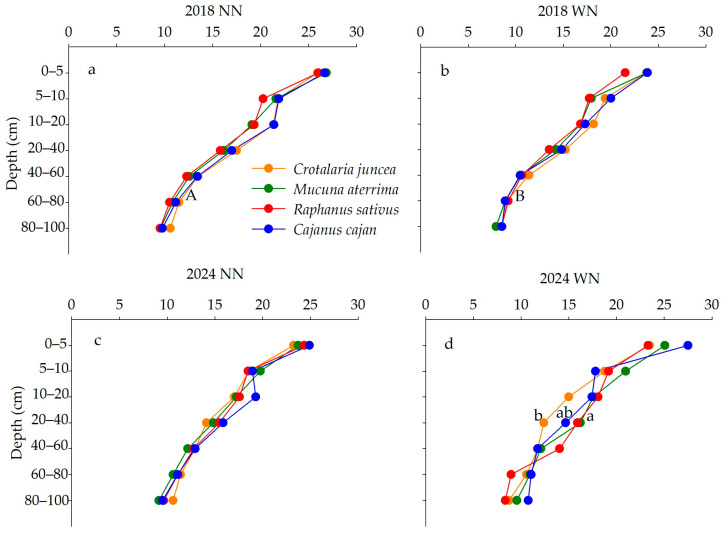
Total soil C content (g kg^−1^) in maize phase (2018) without N (**a**) and with N fertilization (**b**); and end of soybean phase (2024) (NN) without N (**c**) and (WN) with N (**d**) (fertilization applied in maize phase) followed by CCs. Lowercase letters indicate statistical differences between plot treatments (cover crops); uppercase letters represent statistical differences between subplot treatments (N management—NN and WN) by Tukey’s test (*p* < 0.05).

**Figure 2 plants-15-00090-f002:**
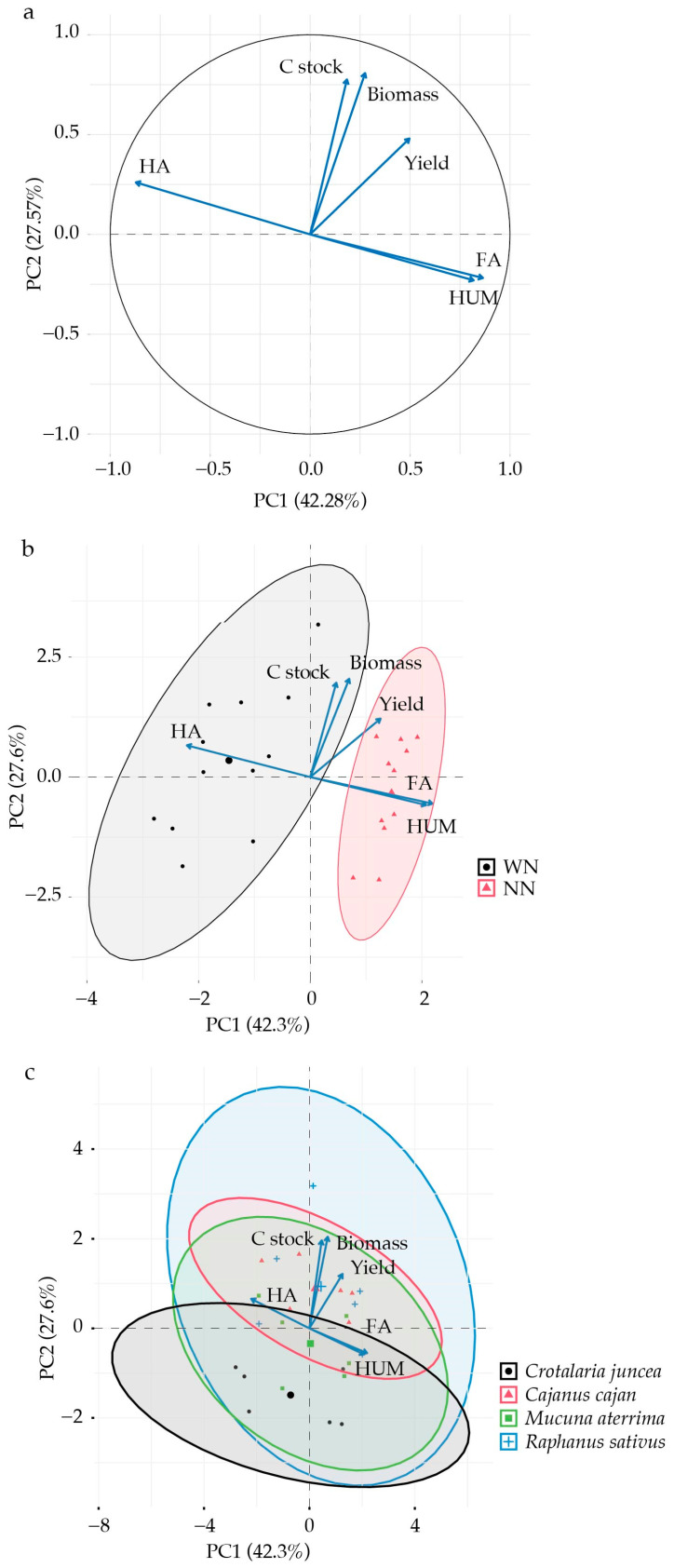
Principal component analysis related to soil C stocks, soil organic matter fractions (FA, HA, and HUM), biomass, and yield. (**a**) Eigenvector correlation circle of variables. (**b**) Distribution of treatments with and without N topdressing. (**c**) Correlation with crop performance in 2024.

**Figure 3 plants-15-00090-f003:**
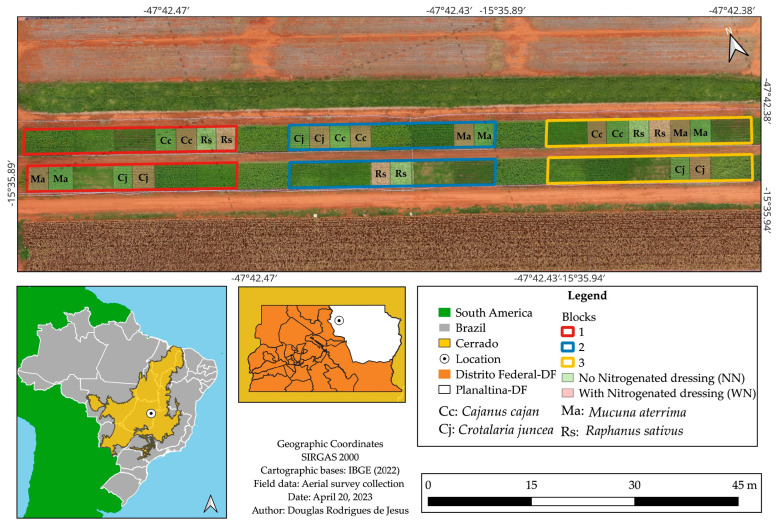
Experimental area at Embrapa Cerrados, in Planaltina, Distrito Federal, Brazil.

**Figure 4 plants-15-00090-f004:**
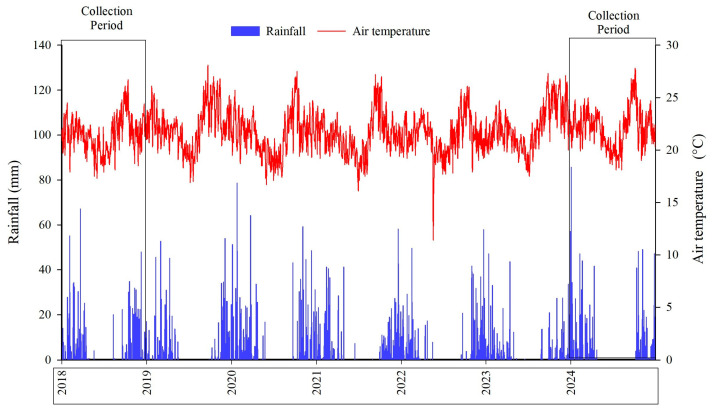
Rainfall (mm) and mean air temperature (°C) from January 2018 to December 2024, Planaltina, Distrito Federal, Brazil.

**Figure 5 plants-15-00090-f005:**
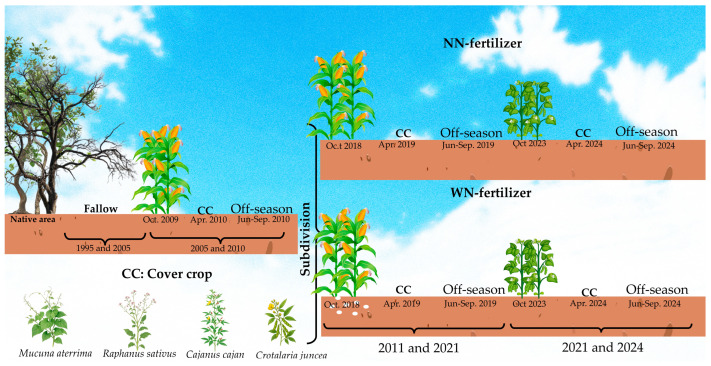
Land-use sequence of the experimental area, 1995–2024.

**Table 1 plants-15-00090-t001:** Mean (±standard deviation) of C and N input from CC biomass in 2017.

Cover Crops	NN	WN	NN	WN
	2017 (kg ha^−1^)			
	Carbon Input		Nitrogen Input	
*Mucuna aterrima*	264.36 (±5.36) aB	381.12 (±142.13) aA	10.58 (±0.85) aB	17.06 (±6.47) aA
*Raphanus sativus*	265.37 (±9.84) aA	285.41 (±12.82) aA	10.23 (±0.79) aA	12.29 (±0.79) aA
*Cajanus cajan*	263.91 (±2.92) aA	289.09 (±6.64) aA	10.9 (±0.66) aA	12.5 (±0.50) aA
*Crotalaria juncea*	267.67 (±3.71) aA	284.04 (±19.56) aA	1.86 (±0.43) aA	12.48 (±0.89) aA

Different lowercase letters in a column indicate statistical differences between plot treatments (cover crops), while uppercase letters in a row represent statistical differences between subplot treatments (N management—NN and WN) (Tukey’s test at 5% probability).

**Table 2 plants-15-00090-t002:** Mean (±standard deviation) of C and N input from CC biomass in 2024.

Cover Crops	NN	WN	NN	WN
	2023 (kg ha^−1^)			
	Carbon Input		Nitrogen Input	
*Mucuna aterrima*	585.16 (±42.20) abA	600.40 (±100.40) abA	38.57 (±1.71) abA	40.87 (±6.74) aA
*Raphanus sativus*	530.71 (±131.77) bA	729.54 (±292.56) aA	28.37 (±7.56) abA	40.11(±17.84) aA
*Cajanus cajan*	905.26 (±40.52) aA	695.22 (±102.12) aA	52.66 (±9.99) aA	39.34 (±7.13) abA
*Crotalaria juncea*	343.95 (±59.48) bA	262.23 (±52.55) bA	21.34 (±5.14) bA	14.49 (±2.93) bA

Different lowercase letters in a column indicate statistical differences between plot treatments (cover crops), while uppercase letters in a row represent statistical differences between subplot treatments (N management—NN and WN) (Tukey’s test at 5% probability).

**Table 3 plants-15-00090-t003:** Mean (±standard deviation) of soil carbon stocks (Mg ha^−1^) at 0–100 cm.

Cover Crops	Soil Carbon Stock (Mg ha^−1^)	
	NN	WN
	2018	
*Mucuna aterrima*	129.40 (±1.50) aA^a^	132.76 (±2.84) aA^a^
*Raphanus sativus*	126.88 (±2.10) aA^a^	133.31(±3.11) aA^a^
*Cajanus cajan*	135.79 (±8.26) aA^a^	133.18 (±2.80) aA^a^
*Crotalaria juncea*	138.51 (±9.99) aA^a^	135.92 (±4.09) aA^a^
	2024	
*Mucuna aterrima*	121.00 (±2.09) aA^b^	111.98 (±7.24) aA^b^
*Raphanus sativus*	125.05 (±3.44) aA^b^	122.38 (±4.63) aA^b^
*Cajanus cajan*	127.91 (±10.58) aA^b^	124.28 (±15.23) aA^b^
*Crotalaria juncea*	124.04 (±6.85) aA^b^	126.23 (±3.17) aA^b^

Different lowercase letters in a column indicate statistical differences between plot treatments (cover crops), while uppercase letters in a column represent statistical differences between subplot treatments (N rates—NN and WN) (Tukey’s test at 5% probability).

**Table 4 plants-15-00090-t004:** Mean (±standard deviation) soil carbon fractions (g kg^−1^) at different soil depths (0–10, 10–20, and 20–40 cm).

Cover Crops	^1^FA	^2^HA	^3^HUM
	NN		
	0–10 cm		
*Mucuna aterrima*	5.82 (±0.94) aA	1.85 (±0.21) aA	9.05 (±1.12) aA
*Raphanus sativus*	6.57 (±0.14) aA	1.79 (±0.05) aA	9.01 (±0.66) aA
*Cajanus cajan*	6.35 (±0.16) aA	1.71 (±0.13) aA	9.06 (±0.63) aA
*Crotalaria juncea*	6.16 (±0.43) aA	1.77 (±0.16) aA	8.91 (±0.75) aA
	10–20 cm		
*Mucuna aterrima*	5.38 (±079) aA	1.10 (±0.28) aB	7.75 (±0.78) aA
*Raphanus sativus*	6.37 (±0.22) aA	1.26 (±0.07) aB	8.46 (±1.07) aA
*Cajanus cajan*	6.02 (±0.32) aA	1.56 (±0.22) aA	8.48 (±1.48) aA
*Crotalaria juncea*	6.00 (±0.29) aA	1.27 (±0.19) aB	8.70 (±0.53) aA
	20–40 cm		
*Mucuna aterrima*	5.91 (±0.11) aA	1.02 (±0.21) aB	7.62 (±0.82) aA
*Raphanus sativus*	6.24 (±0.14) aA	1.20 (±0.03) aB	8.47 (±0.51) aA
*Cajanus cajan*	5.95 (±0.22) aA	1.22 (±0.05) aA	7.70 (±0.61) aA
*Crotalaria juncea*	5.88 (±0.25) aA	1.11 (±0.01) aB	7.90 (±0.06) aA
	WN		
	0–10 cm		
*Mucuna aterrima*	3.65 (±0.71) bB	2.00 (±0.19) aA	7.73 (±1.02) aA
*Raphanus sativus*	3.77 (±0.58) abB	2.27 (±0.14) aA	8.38 (±0.49) aA
*Cajanus cajan*	5.30 (±0.48) aA	2.06 (±0.13) aA	6.51 (±1.40) aA
*Crotalaria juncea*	4.13 (±0.16) abB	2.21 (±0.14) aA	6.74 (±0.33) aA
	10–20 cm		
*Mucuna aterrima*	3.88 (±0.50) aA	2.03 (±0.13) aA	7.14 (±1.16) aA
*Raphanus sativus*	3.63 (±0.36) aA	1.90 (±0.15) aA	7.06 (±1.40) aA
*Cajanus cajan*	5.30 (±0.48) aA	2.05 (±0.03) aA	6.26 (±1.29) aA
*Crotalaria juncea*	4.41 (±1.09) aA	2.07 (±0.04) aA	6.16 (±0.46) aA
	20–40 cm		
*Mucuna aterrima*	4.04 (±0.67) aB	1.78 (±0.24) aA	6.29 (±1.45) aA
*Raphanus sativus*	3.59 (±0.57) aB	1.79 (±0.05) aA	6.09 (±1.03) aA
*Cajanus cajan*	4.73 (±0.30) aA	1.87 (±0.08) aA	4.72 (±0.50) aB
*Crotalaria juncea*	3.39 (±1.01) aB	1.67 (±0.13) aA	5.98 (±1.51) aA

Different lowercase letters in a column indicate statistical differences between plot treatments (cover crops), while uppercase letters in a row represent statistical differences between subplot treatments (N managements—NN or WN) (Tukey’s test at 5% probability). ^1^FA = fulvic acid; ^2^HA = humic acid; ^3^HUM = humin.

**Table 5 plants-15-00090-t005:** Mean (±standard deviation) variations in soil C stocks (ΔC) from 2018 to 2024 under different CC treatments and N managements.

Cover Crops	NN	WN
	2018–2024	
*Mucuna aterrima*	−8.4 (±0.98) bA	−20.78 (±7.86) bB
*Raphanus sativus*	−1.82 (±1.34) aA	−10.93 (±1.78) abB
*Cajanus cajan*	−7.87 (±3.17) bcA	−18.91 (±0.99) bA
*Crotalaria juncea*	−22.41 (±1.72) cA	−9.69 (±1.18) aA

Different lowercase letters in a column indicate statistical differences between plot treatments (cover crops), while uppercase letters in a row represent statistical differences between subplot treatments (N managements—NN or WN) (Tukey’s test at 5% probability).

**Table 6 plants-15-00090-t006:** Mean (±standard deviation) maize and soybean yields following cover crops in 2018 and 2024.

Cover Crops	NN	WN
	Maize Yield (kg ha^−1^) 2018/2019	
*Mucuna aterrima*	7831 (±554) aB	11,326 (±959) aA
*Raphanus sativus*	6153 (±552) bB	10,916 (±110) aA
*Cajanus cajan*	7640 (±533) abB	11,135 (±635) aA
*Crotalaria juncea*	8153 (±442) aB	11,258 (±502) aA
	Soybean yield (kg ha^−1^) 2023/2024	
*Mucuna aterrima*	5153 (±157) aA	4938 (±19) aA
*Raphanus sativus*	5044 (±50) aA	5041 (±168) aA
*Cajanus cajan*	4903 (±41) aA	5094 (±373) aA
*Crotalaria juncea*	4880 (±107) aA	4405 (±67) aA

Different lowercase letters in a column indicate statistical differences between plot treatments (cover crops), while uppercase letters in a row represent statistical differences between subplot treatments (N managements NN and WN) (Tukey’s test at 5% probability).

**Table 7 plants-15-00090-t007:** Mean (±standard deviation) of cover crop dry matter production in 2017 and 2023.

Cover Crops	NN	WN
	2017 (kg ha^−1^)	
*Mucuna aterrima*	1125 (±23) aA	1209 (±23) aA
*Raphanus sativus*	1129 (±42) aA	1216 (±55) aA
*Cajanus cajan*	1123 (±12) aB	1230 (±28) aA
*Crotalaria juncea*	1139 (±16) aA	1209 (±83) aA
	2023 (kg ha^−1^)	
*Mucuna aterrima*	1245 (±128) abA	1277 (±213) abA
*Raphanus sativus*	1437 (±352) abA	1552 (±622) aA
*Cajanus cajan*	1618 (±513) aA	1479 (±217) aA
*Crotalaria juncea*	732 (±127) bA	558 (±112) bA

Different lowercase letters in a column indicate statistical differences between plot treatments (cover crops), while uppercase letters in a row represent statistical differences between subplot treatments (N managements NN and WN) (Tukey’s test at 5% probability).

**Table 8 plants-15-00090-t008:** Soil chemical properties of the experimental area at Embrapa Cerrados, in Planaltina-Distrito Federal, Brazil.

Variables	Values	Units
pH	6.0	-
OM	21.7 ^a^	g kg^−1^
P	0.9	mg kg^−1^
Al^3+^	0.1	cmol_c_ kg^−1^
Ca^2+^ + Mg^2+^	2.9	cmol_c_ kg^−1^
K^+^	0.1	cmol_c_ kg^−1^
Fine sand	258.0	g kg^−1^
Coarse sand	76.7	g kg^−1^
Silt	101.8	g kg^−1^
Clay	563.5	g kg^−1^

pH: in water; OM: organic matter; P: Mehlich^−1^; Al: exchangeable; Ca: exchangeable calcium; Mg: magnesium; K: exchangeable potassium. ^a^ estimated by multiplying organic C × 1.724 (van Bemmelen factor).

## Data Availability

The data presented in this study are available upon request from the corresponding authors.

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
