# Peer review of "Soil Carbon and Organic Matter Fractions Under Nitrogen Management in a Maize–Soybean–Cover Crop System in the Cerrado"

_plants, 2025, doi:10.3390/plants15010090_

Round 1
Reviewer 1 Report
Comments and Suggestions for Authors
Introduction
The introduction contains too much general background. Please focus on the information that directly motivates this field study. It is fine to keep one or two paragraphs on the importance of soil-C storage and forms, and on the benefits of cover crops (CCs) and no-tillage (NTS), but avoid a long review of well-known material. Please structure the introduction so that the reader can see the knowledge gap your study addresses and why this experiment is needed. The storyline should explain why the hypothesis “maize–soybean rotation under NTS with CCs and N management during the maize phase (with and without top-dressed N) enhances crop yield, increases system dry matter, and affects SOM fractions and soil-C stocks” is important. As written, it appears expected; please clarify the novelty and necessity of testing it.
L49: “for straw formation in grain production” is unclear. Do you mean the cover crop or the main crop? Please split into separate sentences, each with a single topic.
Materials and Methods
Because the CC experiment began in 2011, baseline soil data from that time would be valuable for comparison. If not available, please discuss this limitation explicitly. Also please provide a timeline (table or figure) showing all crop and CC planting, management, and harvest dates for the entire experimental period.
L347: Delete “and harvested in March,” since the following information concerns seeding, not harvest.
L350: Provide the product name and company for each fertilizer used.
L359: Paragraph indentation is missing. Please fix formatting.
L360: State the harvest timing for each species. Also describe management after the May harvest until the next seeding (for example, weed control).
L374: Paragraph indentation is missing. Also, were bulk density measurements taken in the plots? If not, please state this and explain why and how that can affect the calculation.
L388: Define CT and DS.
Statistics:
Year is not a sub-subplot but a repeated measure on the same experimental unit. Also, please explain the rationale for using PCA: what question it addresses, which variables were included, how data were scaled or standardized, and how components were interpreted.
Results
If no statistically significant differences were detected, please remove the letters from Table 1.
Please report the total C and N inputs from each cover crop, by year and treatment.
Discussion and Conclusions
Please interpret the findings in light of the motivation stated in the Introduction (why this field study was needed and what gap it aimed to fill). As written, the Discussion mainly restates results and describes related past studies; the reader is left uncertain about the implications (“so what?”).
I recommend structuring the Discussion around the hypothesis and gap:
- Restate the specific hypothesis/question tested in this experiment.
- State whether each key result supports or challenges it.
- Explain the mechanism or plausible rationale for the observed patterns.
- Position the findings vs. prior literature (where do you confirm, extend, or contradict earlier work? what is the specific contribution of this work?).
- Implications: what do these results mean for practice (choice of CCs, N management under NTS), monitoring, etc?
The Conclusion should not be a summary of results. Please include a clear, one-sentence answer to the research question, and a take-home implication for management under NTS with CCs and a concrete next step.
Author Response
Comments and Suggestions for Authors
Introduction
The introduction contains too much general background. Please focus on the information that
directly motivates this field study. It is fine to keep one or two paragraphs on the importance of
soil-C storage and forms, and on the benefits of cover crops (CCs) and no-tillage (NTS), but
avoid a long review of well-known material. Please structure the introduction so that the
reader can see the knowledge gap your study addresses and why this experiment is needed.
The storyline should explain why the hypothesis “maize–soybean rotation under NTS with CCs
and N management during the maize phase (with and without top-dressed N) enhances crop
yield, increases system dry matter, and affects SOM fractions and soil-C stocks” is important. As
written, it appears expected; please clarify the novelty and necessity of testing it.
L49: “for straw formation in grain production” is unclear. Do you mean the cover crop or the
main crop? Please split into separate sentences, each with a single topic. We adjust as requested:
Changes in land use and land cover affect the carbon (C) cycle and its storage in soils [1],
and consequently, greenhouse gas (GHG) emissions [2,3]. It is estimated that soils contain
approximately 24,000 Gt of organic C, accounting for more than 99% of the C accumulated in
terrestrial biomass [2], whereas plant biomass stocks above-ground and belowground (roots)
contain about 2,400 Gt of C. Therefore, soil organic C stocks are highly significant, as they store
more C than the atmosphere and plant bio-mass [4,5].
Fertilization, particularly nitrogen (N) fertilization, applied to achieve the high crop yield
results in increased biomass input and may enhance soil C stocks [6,7]. Ac-cording to the
Yes
( )
(x)
( )
( )
( )
( )
literature, maize exhibits low nitrogen use efficiency, especially in tropical soils, with a maximum
uptake of only about 50% of the N applied as fertilizer [8–10].
Soybean and maize crops are of great importance to Brazil, which ranks as the world’s
second-largest food exporter [11–13]. According to the National Supply Company (CONAB),
Brazil is the world’s leading soybean producer, and the projected cultivated areas for soybean
and maize in the 2024/25 season are 47.36 and 21.01 million hectares, respectively [14]. Crop
rotation and the use of cover crops (CCs) can improve soil health and quality and, consequently,
enhance the productivity of these crops [15], thereby helping to mitigate soil degradation and
reduce the expansion of the agricultural frontier in the Cerrado biome [16]. In this context, these
practices also stimulate soil organism activity [17,18] and increase soil C stocks [19–21], through
biomass inputs from both aboveground and root systems, which generally extend into the deep
er soil layers [18,22].
The use of CCs aims to increase nutrient supply (Sousa et al., 2025) for subsequent crops,
particularly N [23,24], through biological nitrogen fixation (BNF) and the decomposition of plant
residues [20,25], which contribute to the accumulation of chemical fractions of soil organic
matter (SOM), such as fulvic acids (FA), humic acids (HA), and humin (HUM) [26]. HUM is the
insoluble fraction of humic substances (HSs); HA dissolves under alkaline conditions, whereas FA
is soluble under both acidic and alkaline conditions [27]. The SOM fractions are among the main
indicators of soil quality, particularly in highly weathered Oxisols [19].
The HSs have emerged as multifunctional natural catalysts in sustainable agriculture.
They interact with ions to form complexes of varying stability and structural characteristics,
creating new perspectives for improving soil health, plant productivity, and environmental
resilience [28,29]. All these properties can indicate the quality of soil management in agricultural
areas [31]. Their complex molecular structure represents an abundant source of carbon (C) and
energy for beneficial soil microorganisms such as bacteria, fungi, and actinomycetes [32]. Thus,
humic fractions tend to increase particle cohesion and aggregate stability and are associated
with aggregate-size distribution and soil carbon conservation [33,34].
Soil C contents and stocks may increase or decrease depending on the combination of
crop rotations and cover crops grown in succession. In the present study, soil C stocks increased
during the maize phase but declined significantly after the transition to soybean in 2021, except
in the treatments with sorghum (Sorghum bicolor) and wheat (Triticum aestivum) [30]. Based on
the hypothesis that the maize–soybean transition under no-tillage (NT) with the use of CCs
promotes greater N supply efficiency and enhances biomass production within the agricultural
system, consequently in-creasing soil C stocks, this study aimed to evaluate soil C contents and
stocks in 2018 and 2024, assess the delta (ΔC), and determine maize and soybean productivity
as well as CC biomass/dry matter production in the same years. Additionally, the study evalu
ated the humic fractions of soil organic matter (SOM) in 2024 in a long-term experiment under
treatments with and without N fertilizer applied as topdressing for the maize crop.
Materials and Methods
Because the CC experiment began in 2011, baseline soil data from that time would be valuable
for comparison. If not available, please discuss this limitation explicitly. Also please provide a
t
imeline (table or figure) showing all crop and CC planting, management, and harvest dates for
the entire experimental period.
We included a figure in the manuscript illustrating the experimental layout and management
practices.
L347: Delete “and harvested in March,” since the following information concerns seeding, not
harvest.
We have corrected it in the text.
L350: Provide the product name and company for each fertilizer used.
Each year, a different fertilizer brand was used, depending on product availability in the market.
L359: Paragraph indentation is missing. Please fix formatting.
We have corrected it in the text.
L360: State the harvest timing for each species. Also describe management after the May
harvest until the next seeding (for example, weed control).
We have corrected it in the text.
L374: Paragraph indentation is missing. Also, were bulk density measurements taken in the
plots? If not, please state this and explain why and how that can affect the calculation.
Soil bulk density was not measured in the plots due to the small plot size and the large volume
of soil sampled from trenches.
L388: Define CT and DS.
We have corrected it in the text.
Statistics:
Year is not a sub-subplot but a repeated measure on the same experimental unit. Also, please
explain the rationale for using PCA: what question it addresses, which variables were included,
how data were scaled or standardized, and how components were interpreted.
First, we thank the reviewer for the comment that pointed out the need to improve the
methodological description. In our original submission, the explanation of the statistical
methodology was not sufficiently clear regarding the role of Year. Although our analyses were
conducted using a mixed-model structure in which Year was considered as a repeated measure
on the same experimental unit, we inadvertently described it as a “sub-subplot” in the text,
which could lead to misunderstandings. We have now clarified this point in the revised
manuscript.
The methodology explicitly states that Year was modeled as a repeated measure, taking into
account the temporal dependence among observations from the same experimental unit across
years (Piepho et al., 2004; Pagliari et al., 2022). We also included the mathematical structure of
the linear mixed model to provide transparency regarding the specification of fixed effects,
random effects, and the covariance structure for repeated measures.
Once again, we sincerely thank the reviewer for highlighting the lack of clarity in our
methodological description.
In addition, a Principal Component Analysis (PCA) was conducted to explore multivariate
patterns and relationships among soil attributes (C content and fractions) and crop responses
(yield and biomass). The rationale for using PCA was to integrate multiple correlated variables
into synthetic axes (principal components), thereby reducing dimensionality and highlighting the
main gradients of variation associated with management practices. Prior to the analysis,
variables were centered and scaled to unit variance to avoid bias due to differences in
measurement units. The components were interpreted based on variable loadings, with PC1 and
PC2 representing the main directions of variation. These multivariate patterns allowed us to
identify which treatments were associated with higher soil C fractions, greater biomass, or
improved yield.
Results
If no statistically significant differences were detected, please remove the letters from Table 1.
There was a difference between “Years“.
Please report the total C and N inputs from each cover crop, by year and treatment.
2.1. C and N Input via Cover Crops Biomass
No significant differences were observed among treatments for C and N inputs via cover crop
biomass in 2017. In the comparison between subplots (WN and NN), Mucuna aterrima showed
the highest C and N inputs when N topdressing was applied to the maize crop (Table 1).
In 2023, influenced by soybean as the main crop, C inputs were higher in the treatments
with Raphanus sativus and Cajanus cajan compared to Crotalaria juncea in the WN subplots
(residual effect of maize cultivation). In the NN subplots, C input was higher in the Cajanus cajan
treatment than in Raphanus sativus and Crotalaria juncea. No differences were observed
between N management practices. For N input, Mucuna aterrima and Raphanus sativus showed
the highest values compared to Crotalaria juncea in subplots with N topdressing on maize. In the
NN subplots, Cajanus cajan differed from Crotalaria juncea. Again, no differences were observed
between N management practices (Table 2).
For the Mucuna aterrima treatment in 2017, the N stock in biomass allowed pre
decomposition assessments to capture increases in total N (Carvalho et al., 2024). The structural
composition of residues, particularly high lignin content, slows decomposition and
mineralization, thereby altering the temporal availability of N and enhancing its re-tention in the
soil particulate fractions (Sousa et al., 2019). Moreover, the presence of inor-ganic N in the
rhizosphere modulates biological N fixation, reducing symbiotic investment and generating
variations in the source of accumulated N (Bloch et al., 2020). Additionally, the effect of cover
crops (CCs) on C and N stocks is manifested over the long term, whereas short cycles between
incorporation and assessment prioritize the direct effect of fertilizer on maize, at the expense of
the residual benefits of cover crops (Vilches-Ortega et al., 2022; Rocha et al., 2020).
In 2023, C input was higher in Raphanus sativus and Cajanus cajan compared to Crot
alaria juncea, due to the C:N ratio and lignin content of the residues; it should be noted that
nitrogen fertilization did not significantly affect C input (Carvalho et al., 2024). Regarding N input,
Mucuna aterrima and Raphanus sativus showed the highest values in the presence of N fertilizer,
whereas Cajanus cajan stood out in the absence of N fertilizer, confirming its high symbiotic
f
ixation capacity. These results demonstrate that N cycling by CCs is stable and highlight Cajanus
cajan as a strategic specie in low-fertilizer input systems (Berriel & Perdomo, 2023).
Discussion and Conclusions
Please interpret the findings in light of the motivation stated in the Introduction (why this field
study was needed and what gap it aimed to fill). As written, the Discussion mainly restates
results and describes related past studies; the reader is left uncertain about the implications
(“so what?”).
I recommend structuring the Discussion around the hypothesis and gap:
1. Restate the specific hypothesis/question tested in this experiment.
2. State whether each key result supports or challenges it.
3. Explain the mechanism or plausible rationale for the observed patterns.
4. Position the findings vs. prior literature (where do you confirm, extend, or contradict
earlier work? what is the specific contribution of this work?).
5. Implications: what do these results mean for practice (choice of CCs, N management
under NTS), monitoring, etc?
The Conclusion should not be a summary of results. Please include a clear, one-sentence
answer to the research question, and a take-home implication for management under NTS with
CCs and a concrete next step.
We worked on some points in the discussion and conclusions to improve understanding,
as suggested by the reviewer.

Reviewer 2 Report
Comments and Suggestions for Authors
The Introduction chapter is almost fine. Focus on the stability of fulvic and humic acids you mentioned. Many authors question it (I personally don't).
The Results and Discussion chapters can be improved, but I don't see the main problem of the article in them.
I need to explain (specify in the article):
1) Lines 330-334: I think it would be appropriate to state not only the years in which the experiment was conducted, but also the long-term averages.
2) Line 336: What does "organic matter (OM), 21.7 g kg⁻¹" mean? State it as the organic carbon content.
3) Line 336: Mehlich-1? Overall, I think that the data in lines 329-338 would be appropriate to present in a table.
In general, I find the chapter describing the methodology confusing. I think it would be a good idea to rewrite it so that the reader doesn't get lost in it. On the contrary, they should immediately understand what experiments were performed.
Although the Conclusion is summarized in specific points, it sounds very general. Replace these statements with specific results.
Resume: Please significantly improve/simplify the description of the methodology. It is also important that the reader understands the results of your research. So rewrite both the Conclusion and the Abstract. I don't think it is necessary to write that the cultivation technologies you have proven can help mitigate greenhouse gas emissions. In my opinion, this is only a temporary effect.
Author Response
The Introduction chapter is almost fine. Focus on the stability of fulvic and humic acids you
mentioned. Many authors question it (I personally don't).
The use of CCs aims to increase nutrient supply (Sousa et al., 2025) for subsequent crops,
particularly N [23,24], through biological nitrogen fixation (BNF) and the decomposition of plant
residues [20,25], which contribute to the accumulation of chemical fractions of soil organic
matter (SOM), such as fulvic acids (FA), humic acids (HA), and humin (HUM) [26]. HUM is the
insoluble fraction of humic substances (HSs); HA dissolves under alkaline conditions, whereas FA
is soluble under both acidic and alkaline conditions [27]. The SOM fractions are among the main
indicators of soil quality, particularly in highly weathered Oxisols [19].
The HSs have emerged as multifunctional natural catalysts in sustainable agriculture.
They interact with ions to form complexes of varying stability and structural characteristics,
creating new perspectives for improving soil health, plant productivity, and environmental
resilience [28,29]. All these properties can indicate the quality of soil management in agricultural
areas [31]. Their complex molecular structure represents an abundant source of carbon (C) and
energy for beneficial soil microorganisms such as bacteria, fungi, and actinomycetes [32]. Thus,
humic fractions tend to increase particle cohesion and aggregate stability and are associated
with aggregate-size distribution and soil carbon conservation [33,34].
The Results and Discussion chapters can be improved, but I don't see the main problem of the
article in them.
Thank you for your feedback. We have revised the Results and Discussion sections to improve
clarity.
I need to explain (specify in the article):
1) Lines 330-334: I think it would be appropriate to state not only the years in which the
experiment was conducted, but also the long-term averages.
During the study period, the average annual rainfall was 1.187 mm and the average air
temperature was 21.73.
2) Line 336: What does "organic matter (OM), 21.7 g kg⁻¹ mean? State it as the organic carbon
content.
We kept the unit g kg⁻¹ to maintain consistency with international standards.
3) Line 336: Mehlich-1? Overall, I think that the data in lines 329-338 would be appropriate to
present in a table.
We added the reference Sparks (1996) as the manual used for soil analyses. The Soil chemical
characterization are presented in Table 8.
In general, I find the chapter describing the methodology confusing. I think it would be a good
idea to rewrite it so that the reader doesn't get lost in it. On the contrary, they should
immediately understand what experiments were performed.
We made several adjustments to improve reader comprehension.
Although the Conclusion is summarized in specific points, it sounds very general. Replace these
statements with specific results.
Thank you for your observation. We have included some more specific points as recommended:
Resume: Please significantly improve/simplify the description of the methodology. It is also
important that the reader understands the results of your research. So rewrite both the
Conclusion and the Abstract. I don't think it is necessary to write that the cultivation technologies
you have proven can help mitigate greenhouse gas emissions. In my opinion, this is only a
temporary effect
We followed the reviewers’ recommendations by simplifying the methodology section and using
more direct language. Minor modifications were also made to the summary and conclusion to
improve clarity.

Round 2
Reviewer 1 Report
Comments and Suggestions for Authors
Thank you for the resubmission. As this is the second round, I was hoping to focus on the scientific substance. However, the manuscript still contains pervasive typographical and language errors (including in the title), to the extent that I am not able to read beyond the first page with confidence. Examples include malformed words such as “SOMOrganic,” “MAagamenemtMam-agement,” “relevanceimportance,” “isranks,” and “thirdsecond,” as well as numerous punctuation and grammar issues.
Before I can provide a full scientific assessment, I strongly recommend a thorough language revision—ideally by a professional scientific editor or a proficient English-speaking colleague—followed by resubmission. Below I list a few illustrative, line-specific notes from the title and abstract to help guide the revision.
Specific comments (up to where I could read)
L29: “When comparing soil C stocks during the maize phase, higher mean values were observed (p < 0.05)”
Please specify the explicit comparator. Higher than what? (e.g., maize phase vs. cover-crop phase?)
L33: “with direct implications for greenhouse gas (GHG) mitigation”
Based on the previous sentences alone, the causal link to GHG mitigation is not established. If you assert “Thus…GHG mitigation,” briefly explain the logical link to GHG mitigation.
L34: Missing period at the end of the sentence.
L34–35: The final sentence should be a conclusion but its logical connection to the preceding results is unclear. Please revise the abstract’s flow so that conclusions follow directly from the stated findings.
L46: The topic sentence discusses N fertilization and soil C stocks, but the following sentence shifts to maize N-use efficiency without explaining the linkage. As a two-sentence paragraph, this feels disjointed.
Comments on the Quality of English LanguageThe manuscript still contains pervasive typographical and language errors (including in the title), to the extent that I am not able to read beyond the first page with confidence.
Author Response
Dear Reviewer,
We are very thankful for the clear and objective comments made by the reviewers. We agree with all the points raised by the reviewers and have done our best to address them properly. We believe that the revised manuscript has improved significantly as a result. Please see our detailed responses (in blue) below.
Reviewers' comments:
Reviewer #1:
Comments and Suggestions for Authors
Thank you for the resubmission. As this is the second round, I was hoping to focus on the scientific substance. However, the manuscript still contains pervasive typographical and language errors (including in the title), to the extent that I am not able to read beyond the first page with confidence. Examples include malformed words such as “SOMOrganic,” “MAagamenemtMam-agement,” “relevanceimportance,” “isranks,” and “thirdsecond,” as well as numerous punctuation and grammar issues.
Answer: We sincerely apologize for the language and typographical errors. We have carefully revised the text and corrected all identified errors.
Before I can provide a full scientific assessment, I strongly recommend a thorough language revision—ideally by a professional scientific editor or a proficient English-speaking colleague—followed by resubmission. Below I list a few illustrative, line-specific notes from the title and abstract to help guide the revision.
Answer: We appreciated the constructive comments.
Specific comments (up to where I could read)
Reviewer: L29: “When comparing soil C stocks during the maize phase, higher mean values were observed (p < 0.05)”
Please specify the explicit comparator. Higher than what? (e.g., maize phase vs. cover-crop phase?)
Answer: Thank you for the comment. It has been corrected.
Reviewer: L33: “with direct implications for greenhouse gas (GHG) mitigation”
Based on the previous sentences alone, the causal link to GHG mitigation is not established. If you assert “Thus…GHG mitigation,” briefly explain the logical link to GHG mitigation.
Answer: Thanks for the suggestion. The abstract has been reformulated.
L34: Missing period at the end of the sentence.
Answer: It has been corrected.
L34–35: The final sentence should be a conclusion but its logical connection to the preceding results is unclear. Please revise the abstract’s flow so that conclusions follow directly from the stated findings.
Answer: Thanks for the suggestion. The abstract has been reformulated to meet the reviewers' requests.
L46: The topic sentence discusses N fertilization and soil C stocks, but the following sentence shifts to maize N-use efficiency without explaining the linkage. As a two-sentence paragraph, this feels disjointed.
Answer: Thanks for your thoughtful comment. A sentence has been added to increase the clarity of the paragraph.
Comments on the Quality of English Language
The manuscript still contains pervasive typographical and language errors (including in the title), to the extent that I am not able to read beyond the first page with confidence.
Answer: The manuscript has been carefully revised by a professional language editing service to improve the grammar and readability.
Best regards Reviewer

Reviewer 2 Report
Comments and Suggestions for Authors
Thanks for improving the article. But I still need to clarify what organic matter means on line 445. I understand the units. But how did you get this number (21.7 g kg⁻¹)? Is it organic carbon content? Or Cox*1.724? This is not clear, please clarify in the article.
There are typos in the pdf file that was sent to me due to corrections. I assume this will be fixed in the final version.
Best regards Reviewer
Author Response
Dear Reviwer,
We are very thankful for the clear and objective comments made by the reviewers. We agree with all the points raised by the reviewers and have done our best to address them properly. We believe that the revised manuscript has improved significantly as a result. Please see our detailed responses (in blue) below.
Reviewers' comments:
Reviewer #2:
Comments and Suggestions for Authors
Thanks for improving the article. But I still need to clarify what organic matter means on line 445. I understand the units. But how did you get this number (21.7 g kg⁻¹)? Is it organic carbon content? Or Cox*1.724? This is not clear, please clarify in the article.
Answer: Thanks for your positive comments. We agree with you. The OM value of 21.7 g kg-1 in Table 8 was obtained by multiplying the organic C value by 1.724 (van Bemmelen factor). This information has been added as a footnote to Table 8.
There are typos in the pdf file that was sent to me due to corrections. I assume this will be fixed in the final version.
Answer: Typographical errors have been corrected throughout the text. Moreover, the manuscript has been carefully revised by a professional language editing service to improve the grammar and readability.
Best regards Reviewer

Round 3
Reviewer 1 Report
Comments and Suggestions for Authors
Introduction:
L45–: This paragraph discusses the effect of N fertilization on soil C stock. However, the next paragraph shifts to maize and soybean before substantiating the statement that “the effects of N fertilization on soil C accumulation vary depending on plant species and N-use efficiency.” Please clarify what is known about the relationship between species differences and N-use efficiency, and how these relate to the effect of N fertilization on soil C accumulation.
L50–: This paragraph suddenly switches to Brazil, and midway introduces cover crops. Please ensure a clear, logical paragraph structure. Up to the second paragraph, soil C stock appears to be the central topic; in the third paragraph, the topic shifts abruptly and feels out of place.
L60–: This paragraph discusses cover crops and HA, but the connection to the paper’s main topic is unclear. From the first two paragraphs, readers expect background on soil C accumulation in relation to N inputs. The subsequent discussion of Brazil, cover crops, and HA lacks a clear link to the initial topic. Please make the logical connection explicit.
L77: What does “According to,” refer to? According to which source?
L79–: “Based on the hypothesis”: Where does this hypothesis come from? Is it derived from reference [34]? Why is no-tillage introduced here without prior explanation? What do you mean by “N supply efficiency”? These concepts are not defined earlier. Please ensure that the hypothesis/objectives presented in the final paragraph are firmly supported by the preceding logic of the Introduction.
Without a clear core rationale in the Introduction, the Discussion and Conclusion cannot be developed properly. I will re-evaluate the manuscript once the Introduction has been revised accordingly.
Results:
General: Only in Section 2.1 are WN data shown on the left and NN on the right; elsewhere NN appears first. Please make the ordering consistent throughout.
L92: What is WN? Because M&M comes later, the Results should be interpretable without referring back. Please define all labels/abbreviations in the Results when first used.
L93: What does “highest” mean here? If the comparison is between NN and WN within each cover crop, “highest” across groups is not appropriate.
L97: What do you consider as “treatments”? Is N management not treated as a treatment? The same issue appears in all tables.
L101: “influenced by the main crop soybean” is an interpretation. How do you infer this influence from the data? (without any comparison within the same year).
L106: “maize N topdressings” should be WN. Please use the defined treatment labels consistently. Once an abbreviation is introduced, stick to it throughout (this applies to all Results text to facilitate checking against tables/figures).
L118–119: This type of explanatory phrasing is needed for each result in Section 2.1.
L120–122: I could not follow the statement, and I do not see corresponding evidence in Fig. 1a,b. What does “highest” mean? Also, why introduce “maize–” here? You are comparing 2018 data only; crop rotation itself is not a treatment. Do you also intend to compare maize vs. soybean effects? If so, year effects are confounded, and you cannot simply compare 2018 with 2024 as different “crop rotation” treatments. The same applies to “soybean–” in L123–124.
L137: According to Table 3, C stocks are higher in 2018 than in 2024 (not vice versa).
L150–154: Since the comparison is between NN and WN, “highest” is not appropriate. The same issue occurs at L169.
Please review and correct the remainder of the Results accordingly. Additionally, based on the Introduction, I expected an assessment of N-use efficiency or N-supply efficiency. Where are those results presented?
Materials and Methods:
L368: Do you mean “Jan 2015 to Dec. 2024”? Please correct the dates (currently “Jan 2015 to Dec. 2014”).
Fig. 5: The label “2018” under the maize drawing is confusing and as a whole, difficult to understand. A schematic timeline (cover crops and cash crops with months for sowing/harvest) would be clearer than dense graphics, especially since the text frequently mentions months.
L409: There is a 3-month difference among cover crop species. How did you manage weeding after termination for those with short season cover crops, while other cover crops were still growing?
L420: Was a separate soil sampling conducted for SOM fractions? Was it at the same timing as the main soil sampling?
L432: Please do not use the alphabet letter “x” for multiplication; use the proper mathematical symbol.
L527: There are two occurrences of “(at)ij,” among other inconsistencies that do not match the textual explanations. Please correct all index notations to align with the model description.
L540: Varb is missing.
L540: What exactly do you mean by “treatments” here?
L542: “When applicable” is vague. For which datasets was this method used?
L543: How does this description differ from L540? For which datasets did you use multiple comparisons, and for which did you use ANOVA and Tukey’s test? I also do not see explicit ANOVA results in the Results; I only see multiple-comparison outputs. Did you detect any interaction effects?
Formatting: Please check font size and font usage consistently across the manuscript.
Comments on the Quality of English LanguagePlease check English throughly.
Author Response
Please see attachment, thank you.

Round 4
Reviewer 1 Report
Comments and Suggestions for Authors
L40, L44: There is an extra space after the period.
L40: This sentence is too long and difficult to follow. The phrase “has arisen as nature-based solution” is grammatically incorrect (it should be “has emerged as a nature-based solution”), and the structure with “considering that …” makes the logic unclear. I suggest splitting it into two sentences, for example:
“In this context, increasing soil organic carbon (SOC) storage in croplands has emerged as a nature-based solution for carbon dioxide (CO₂) removal. The global SOC pool to a depth of 1 m is estimated at about 1,500 Pg C, of which croplands account for more than 140 Pg C in the top 30 cm of soil [2].”
L44: Why do you suddenly refer to “tropical regions”? Has the information in this paragraph been about tropical regions in general or about Brazil up to this point (from the first sentence, I assumed that it is about Brazil)? Brazil is not the only place that can be considered “a tropical region.”
L49–: You mention that Brazil is the second largest food exporter and then present detailed figures for the area and yield of soybean and maize. Since you do not compare these values with other crops or countries, the purpose of providing such detailed numbers is not very clear. It may be enough to simply state that soybean and maize are major crops in Brazil, unless you further explain how these specific figures are linked to your research question.
L51: The Cerrado region does not “concentrate” agricultural areas; rather, agricultural land is concentrated in the Cerrado region. Please check subject–verb agreement and phrasing.
I am sorry, but in its current form the text is not yet at a level where I can properly evaluate the content, and I cannot continue to correct the entire paper line by line. Please revise the English thoroughly and ensure that the sentence structure, paragraph structure, and overall logic of the introduction are clear and coherent. I will re-evaluate the manuscript after the English and logical flow have been improved.
Author Response
Dear Reviewer 1,
We are grateful for all your contributions since the first round, and we apologize for the English grammar errors; on many occasions, we did not have sufficient time to submit the text for an English-language review. We have attempted to incorporate all your corrections, and we have also undertaken a new revision with a native English speaker, as indicated in the certificate
L40, L44: There is an extra space after the period.
Corrected in the text
L40: This sentence is too long and difficult to follow. The phrase “has arisen as nature-based solution” is grammatically incorrect (it should be “has emerged as a nature-based solution”), and the structure with “considering that …” makes the logic unclear. I suggest splitting it into two sentences, for example:
“In this context, increasing soil organic carbon (SOC) storage in croplands has emerged as a nature-based solution for carbon dioxide (CO₂) removal. The global SOC pool to a depth of 1 m is estimated at about 1,500 Pg C, of which croplands account for more than 140 Pg C in the top 30 cm of soil [2].”
Corrected in the text
The agriculture, forestry, and other land use (AFOLU) sector accounts for 74% of all Brazilian greenhouse gas (GHG) emissions [1]. This exemplifies how strongly agricultural land-use systems influence soil carbon stocks in tropical regions. The global SOC pool to 1-m depth is estimated at about 1,500 Pg C, of which more than 140 Pg C is found in the top 30 cm of cropland soils [2].
L44: Why do you suddenly refer to “tropical regions”? Has the information in this paragraph been about tropical regions in general or about Brazil up to this point (from the first sentence, I assumed that it is about Brazil)? Brazil is not the only place that can be considered “a tropical region.”
Corrected in the text
The agriculture, forestry, and other land use (AFOLU) sector accounts for 74% of all Brazilian greenhouse gas (GHG) emissions [1]. This exemplifies how strongly agricultural land-use systems influence soil carbon stocks in tropical regions.
L49–: You mention that Brazil is the second largest food exporter and then present detailed figures for the area and yield of soybean and maize. Since you do not compare these values with other crops or countries, the purpose of providing such detailed numbers is not very clear. It may be enough to simply state that soybean and maize are major crops in Brazil, unless you further explain how these specific figures are linked to your research question.
Corrected in the text
As one of the world’s leading agricultural exporters in tropical regions, Brazil depends critically on soybean and maize production [3–5]. According to the Brazilian National Supply Company (CONAB), Brazil remains the world’s leading soybean producer. For the 2024/25 growing season, soybean acreage was projected at 47.35 million hectares, with an expected grain output of 171.5 million tons, while maize was forecast to cover 21.86 mil-lion hectares and yield 139.7 million tons, consolidating Brazil as the third-largest maize producer globally [6]. Approximately 42% of the country’s agricultural land, particularly for grain production, is located in the Cerrado biome [7].
L51: The Cerrado region does not “concentrate” agricultural areas; rather, agricultural land is concentrated in the Cerrado region. Please check subject–verb agreement and phrasing.
Corrected in the text
Approximately 42% of the country’s agricultural land, particularly for grain production, is located in the Cerrado biome [7].
I am sorry, but in its current form the text is not yet at a level where I can properly evaluate the content, and I cannot continue to correct the entire paper line by line. Please revise the English thoroughly and ensure that the sentence structure, paragraph structure, and overall logic of the introduction are clear and coherent. I will re-evaluate the manuscript after the English and logical flow have been improved.
Once again, we apologize for the errors and thank you for your contribution
